# The Flavonoid Cyanidin Shows Immunomodulatory and Broad-Spectrum Antiviral Properties, Including SARS-CoV-2

**DOI:** 10.3390/v15040989

**Published:** 2023-04-18

**Authors:** Josefina Vicente, Martina Benedetti, Paula Martelliti, Luciana Vázquez, María Virginia Gentilini, Freddy Armando Peñaranda Figueredo, Mercedes Soledad Nabaes Jodar, Mariana Viegas, Andrea Alejandra Barquero, Carlos Alberto Bueno

**Affiliations:** 1Laboratorio de Virología, Departamento de Química Biológica, Facultad de Ciencias Exactas y Naturales, Universidad de Buenos Aires, Buenos Aires 1428, Argentina; jvicente@qb.fcen.uba.ar (J.V.); martina_benedetti97@hotmail.com (M.B.); paulamartelliti24_@hotmail.com (P.M.); ffigueredo@qb.fcen.uba.ar (F.A.P.F.); alecab@qb.fcen.uba.ar (A.A.B.); 2Instituto de Química Biológica de la Facultad de Ciencias Exactas y Naturales (IQUIBICEN), CONICET—Universidad de Buenos Aires, Buenos Aires 1428, Argentina; 3Unidad Operativa Centro de Contención Biológica (UOCCB), Administración Nacional de Laboratorios e Institutos de Salud (ANLIS), Buenos Aires 1282, Argentina; lulavazquezuoccb@gmail.com; 4Instituto de Medicina Traslacional, Trasplante y Bioingeniería (IMETTYB)-CONICET, Buenos Aires 1093, Argentina; mvgentilini@gmail.com; 5Consejo Nacional de Investigaciones Científicas y Técnicas (CONICET), Buenos Aires 1425, Argentina; msnabaes@gmail.com (M.S.N.J.); viegasmariana@conicet.gov.ar (M.V.); 6Laboratorio de Virología, Hospital de Niños Ricardo Gutiérrez, Buenos Aires 1417, Argentina

**Keywords:** SARS-CoV-2, antiviral, immunomodulatory, cyanidin, A18, broad-spectrum

## Abstract

New antiviral treatments are needed to deal with the unpredictable emergence of viruses. Furthermore, vaccines and antivirals are only available for just a few viral infections, and antiviral drug resistance is an increasing concern. Cyanidin (a natural product also called A18), a key flavonoid that is present in red berries and other fruits, attenuates the development of several diseases, through its anti-inflammatory effects. Regarding its mechanism of action, A18 was identified as an IL-17A inhibitor, resulting in the attenuation of IL-17A signaling and associated diseases in mice. Importantly, A18 also inhibits the NF-κB signaling pathway in different cell types and conditions in vitro and in vivo. In this study, we report that A18 restricts RSV, HSV-1, canine coronavirus, and SARS-CoV-2 multiplication, indicating a broad-spectrum antiviral activity. We also found that A18 can control cytokine and NF-κB induction in RSV-infected cells independently of its antiviral activity. Furthermore, in mice infected with RSV, A18 not only significantly reduces viral titers in the lungs, but also diminishes lung injury. Thus, these results provide evidence that A18 could be used as a broad-spectrum antiviral and may contribute to the development of novel therapeutic targets to control these viral infections and pathogenesis.

## 1. Introduction

New antiviral treatments are needed to deal with the unpredictable emergence of viruses. To minimize loss of life in future pandemics, we must prospectively produce broad-spectrum antiviral therapies for viruses with pandemic potential [1,2]. Despite many viral diseases causing significant mortality and morbidity in the population, vaccines and antivirals are only available for just a few viral infections, resulting in a great need for effective treatments. In the cases where antivirals or vaccines are available against viral diseases, antiviral drug resistance is an increasing concern, particularly in immunocompromised patient populations, where ongoing viral replication and prolonged drug exposure lead to the selection of resistant strains. Thus, there is an ongoing need for less toxic, but potent, new antiviral drugs that preferably target different aspects of viral replication to reduce the risk of resistance [3].

Most of the current resources of approved antiviral drugs include the so-called direct-acting antiviral agents, which are drugs designed against viral proteins that are essential for infection. However, viruses encode a limited number of proteins, and those suitable as drug targets are only a subset of them. On the other hand, compounds that block viral growth by targeting cellular proteins and pathways instead of the virus itself, so-called host-directed antivirals, are of growing interest because they are less prone to viral evasion and more likely to display broad-spectrum activity [4,5]. Among these, one strategy is to modulate innate defenses using immunomodulatory molecules [4]. Viruses (e.g., HSV-1, RSV, and SARS-CoV-2) trigger host inflammatory responses and modulate cellular signaling networks, such as the NF-kB pathway, which plays a central role in mediating viral-induced inflammation [6,7,8,9].

Cyanidin (a natural product also called A18) is a particular type of anthocyanidin (the sugar-free counterpart of anthocyanins), which is present as a pigment in many red berries and other fruits. Cyanidin and its related compounds have been implicated in attenuating the development of several major diseases, including asthma, diabetes, atherosclerosis, and cancer, by stimulating anti-inflammatory effects [10,11,12,13]. Regarding its mechanism of action, A18 was identified as an IL-17A inhibitor, resulting in the attenuation of IL-17A signaling and associated diseases in mice [14]. Importantly, A18 also inhibits the NF-κB signaling pathway and secretion of proinflammatory cytokines in different cell types and conditions in vitro and in vivo [15,16].

Thus, considering that NF-kB signaling plays an important role in the pathogenesis of many viral infections [6,7,8,9], we deemed it important to investigate the potential of A18 as an antiviral agent. Hence, the first aim of the present study was to evaluate the broad-spectrum antiviral activity of A18 against HSV-1, a double-stranded DNA virus, RSV, a negative-strand RNA virus, and coronaviruses (canine coronavirus and SARS-CoV-2), which are positive-strand RNA viruses. Furthermore, we proposed to analyze its effect on the NF-κB signaling pathway and the production of cytokines in RSV-infected epithelial cells and macrophages. Finally, we intended to evaluate whether these activities were functional in vivo, in a well-characterized model of murine pulmonary RSV infection.

## 2. Materials and Methods

### 2.1. Reagents

Cyanidin chloride (A18; CAS 528-58-5) was obtained from Santa Cruz Biotechnology (Dallas, TX, USA). Pam2CSK4 (TLR2/TLR6 ligand) and poly(I:C)-HMW (TLR 3 ligand) were purchased from InvivoGen (San Diego, CA, USA). The mouse monoclonal antibody anti-gF of RSV was obtained from US Biological Life Sciences (Salem, MA, USA). The rabbit polyclonal antibody anti-Spike of SARS-CoV-2 was obtained from Sigma-Aldrich (St. Louis, MO, USA). Secondary goat anti-mouse FluoroLinkTM CyTM3 and goat anti-rabbit FluoroLinkTMCyTM2 antibodies were purchased from GE Healthcare (Chicago, IL, USA). DAPI was purchased from Sigma-Aldrich (St. Louis, MO, USA).

### 2.2. Cells and Viruses

The human lung carcinoma cell line A549 and the human epidermoid cancer cell line HEp-2 were obtained from Asociación Banco Argentino de Células (ABAC) (Buenos Aires, Argentina) and grown in MEM supplemented with 10% inactivated fetal bovine serum (FBS). Murine macrophage cell line J774A.1 was kindly provided by Dr. Osvaldo Zabal (INTA–Castelar, Buenos Aires, Argentina) and grown in RPMI1640 medium supplemented with 10% FBS. Human corneal-limbal epithelial (HCLE) cells were kindly provided by Dr. Ilene K. Gipson and Dr. Pablo Argueso (The Schepens Eye Research Institute, Harvard Medical School, Boston, MA, USA) and grown in GIBCO keratinocyte serum-free medium. Vero E6 cells were grown in MEM supplemented with 10% FBS. Vero E6 and SARS-CoV-2 Wuham strain were kindly provided by Dr. Jorge Quarleri (INBIRS-Buenos Aires Argentina). Calu-3 cells were kindly provided by Dr. Fernanda Elias (Fundación Pablo Cassará-Buenos Aires, Argentina) and grown in D-MEM supplemented with 10% FBS. Human RSV strains A2 and line 19 were kindly provided by Dr. Laura Talarico (INFANT–Buenos Aires, Argentina). Canine coronavirus and CRFK cells were kindly provided by Dr. Carlos Palacios (Fundación Pablo Cassará-Buenos Aires, Argentina). The KOS strain was chosen as HSV-1 wild-type reference and B2006 and field are HSV-1 thymidine kinase-deficient (TK-) strains of HSV-1. All HSV-1 strains, RSV, canine coronavirus, and SARS-CoV-2 were used and propagated at low multiplicity of infection (moi).

### 2.3. Antiviral Activity

To evaluate the broad-spectrum antiviral activity of A18, HCLE cells were infected with HSV-1 KOS and HSV-1 ACV-resistant strains (B2006 and field strains); HEp-2 and A549 cells were infected with RSV strains line 19 and A2; CRFK cells were infected with canine coronavirus; calu-3 cells were infected with SARS-CoV-2. A virus yield inhibition assay was performed in cells grown in 24-well plates infected with a moi of 0.1. Following 1 h of adsorption at 37 °C, the inoculum was removed, and cells were exposed to the compound over 24 h. Then, supernatants were collected and titrated by plaque assay, and the concentration required to reduce the virus yield by 50% (EC_50_) was calculated. We used acyclovir (ACV) against HSV-1 as a reference drug to validate the antiviral activity assay, as described previously [17].

### 2.4. Cytotoxicity Assay

For cell viability analysis, cells grown in 96 well plates were treated with different concentrations of the compound over 24 h. Mock was considered as 100% of cell viability, which was treated with the same concentration of DMSO as the highest dose of A18 treatment. Cell viability was analyzed by the 3-(4,5-dimethylthiazol-2-yl)-2,5-diphenyl tetrazolium bromide (MTT) (Sigma-Aldrich, St. Louis, MO, USA) assay, following the manufacturer’s instructions. The concentration required to reduce cell viability by 50% (CC_50_) was calculated, as described previously [17].

### 2.5. Adsorption and Penetration Assay

To quantify adsorbed virus, cells were infected together with 25 μg/mL and 50 μg/mL of A18 and incubated for 1 h at 4 °C. Then, the inoculum was discarded, and cells were washed, supplemented with fresh medium, and further incubated for 24 h at 37 °C. Virus yields were collected and titrated by plaque assay. To determine internalized virus, cells were infected and incubated for 1 h at 4 °C. The inoculum was discarded, and cells were washed and treated with 25 μg/mL and 50 μg/mL of A18 for 2 h at 37 °C. Afterwards, cells were washed, and the non-internalized virus was inactivated with citrate buffer (pH 3) for 1 min. Cells were further incubated with fresh medium for 24 h at 37 °C. Virus yields were collected and titrated by plaque assay.

### 2.6. RSV qRT-PCR Assay

The procedure for qRT-PCR assay was performed as described previously [18]. At 6, 15, and 24 h p.i., RNA was extracted with QIAamp^®^ Viral RNA Mini Kit (QIAGEN, Hilden, Germany) according to the manufacturer’s instructions. Then, cDNA synthesis and the qPCR were performed with a primer that recognizes the M gene. RNase P DNA was used as a reference gene. 2^−ΔΔCt^ (fold-change) analysis was established in relation to untreated-virus infected cells at 6 h p.i.

### 2.7. Immunofluorescence Assay (IF) and Semiquantitative IF Analysis

The procedure for IFI was performed as described previously [18]. Cells were fixed with methanol for 10 min at −20 °C, washed with PBS, and stored in PBS at 4 °C until processing. Next, cells were incubated with the primary and secondary antibodies for 30 min at 37 °C and then incubated with DAPI. Microscopy and photography data were obtained using an Olympus IX71 fluorescence microscope. Analysis was performed using Fiji software (version 1.53v) (National Institutes of Health, Bethesda, MD, USA). The percentage of positive cells was expressed as the ratio between viral glycoprotein positive cells and total cells stained with DAPI in at least 3 coverslips, performed in duplicate.

### 2.8. Cytokine Determination

Supernatants harvested from Hep-2, A549, and J774A.1 cells infected with RSV A2 or stimulated with TLR2/6 and TLR3 ligands, and treated or not with A18 (25 mg/mL) and BAY 11-7085 (10 mg/mL), were used to quantify IL-6, IL-8, and TNF-α by ELISA, following manufacturer’s instructions (BD OptEIATM, Becton–Dickinson, Franklin Lakes, NJ, USA).

### 2.9. Pulmonary Infection in Mouse Model

Housing conditions and experimental procedures were conducted with the approval and under the guidelines of the Comisión Institucional de Cuidado y Uso de Animales de Laboratorio (CICUAL) of the Facultad de Ciencias Exactas y Naturales, Universidad de Buenos Aires, Argentina. Female Balb/C mice (6–8 weeks of age) were purchased from Facultad de Veterinaria, Universidad de Buenos Aires, Argentina. They were housed in individually ventilated cages, with a maximum of 6 animals/cage, in an Animal Facility Biosafety Level 3 (ABSL-3) (UOCCB, ANLIS-Malbrán, Buenos Aires, Argentina). The housing conditions for the animals were optimally established, providing environmental enrichment, and with temperature, humidity, and lighting control, with 12 h dark and 12 h light conditions. Cages were changed at least once per week and checked every day to note the condition of the animals. Mice were infected intranasally (i.n.) with 5 × 10^6^ PFU of RSV line 19 and A2 (50 μL) or 25% (*w/v*) sucrose (50 μL) (viral stocks were stored in 25% (*w/v*) sucrose) under isoflurane anesthesia (6 per group). Either 3 mg/kg or 6 mg/kg of A18 (dissolved in DMSO), or a vehicle (DMSO) for the control group, was given 1 h p.i., i.n. or intraperitoneally (i.p.), respectively. Mice further received a daily dose of A18 until day 4 p.i. Body weights were analyzed each day, and mice were sacrificed on day 4 or 8 p.i. Lungs removed on day 4 p.i. were weighed and used to titrate infectious virus by plaque assay in Vero cells. For gene expression studies, RNA was isolated from the right lungs on day 8 p.i. using TRIzol reagent (ThermoFisher Scientific, Waltham, MA, USA). Left lungs from day 8 p.i. were used for histological sectioning and staining after fixation with Bouin solution for 24 h. An index of pathologic changes in hematoxylin and eosin (H&E) slides was analyzed as previously described [19].

### 2.10. Cytokines qRT-PCR

Pulmonary cytokines expression was assessed as described previously [18]. qRT-PCR TRIzol reagent (ThermoFisher Scientific, Waltham, MA, USA) was used to isolate RNA from mouse lungs. Then, reverse transcription was performed using ImProm-II™ Reverse Transcription System A3800 (Promega, Madison, WI, USA), following the manufacturer’s instructions. Detection was performed using the Bio-Rad iQ5 real-time PCR. Results were analyzed as 2^−ΔΔCt^, and β-actin was used for normalization. The following primers were used (forward and reverse):

IL-6 mouse: F:TAGTCCTTCCTACCCCAATTTCC;

R: TTGGTCCTTAGCCACTCCTTC

PrimerBank ID: 13624311a1

IL-4 mouse: F: GGTCTCAACCCCCAGCTAGT;

R: GCCGATGATCTCTCTCAAGTGAT

PrimerBank ID: 10946584a1

IFN-γ mouse: F: ATGAACGCTACACACTGCATC;

R: CCATCCTTTTGCCAGTTCCTC

PrimerBank ID: 33468859a1

IL-17A mouse: F: TTTAACTCCCTTGGCGCAAAA;

R: CTTTCCCTCCGCATTGACAC

PrimerBank ID: 6754324a1

β-actin mouse: F: GTGACGTTGACATCCGTAAAGA;

R: GCCGGACTCATCGTACTCC

PrimerBank ID: 145966868c1

### 2.11. Transfections and Reporter Gene Assays

Hep-2, A549, and J774A.1 cells were transfected with an NF-kB-LUC reporter vector and β-galactosidase control plasmid and, 24 h later, infected with RSV A2 and treated with A18 (25 mg/mL) and BAY 11-7085 (10 mg/mL) during 6 h. The procedure for transfection was performed as described previously [18]. Lipofectamine 2000 reagent (ThermoFisher Scientific, Waltham, MA, USA) was used for transfection assays following the manufacturer’s instructions. The NF-κB-LUC reporter vector and RSV-β-gal plasmid were provided by Dr. Susana Silberstein (Universidad de Buenos Aires, Argentina). Luciferase Assay System E1500 (Promega, Madison, WI, USA) and β-Galactosidase Enzyme Assay System E2000 (Promega, Madison, WI, USA) were used for reporter quantitation following the manufacturer’s instructions.

### 2.12. Statistical Analysis

One-way analysis of variance (ANOVA) followed by Tukey’s multiple comparison tests was used to assess statistical significance, with the software GraphPad Prism 8.3. *p*-value < 0.05 was considered statistically significant.

## 3. Results

### 3.1. Broad-Spectrum Antiviral Activity of A18

A18 significantly reduced infectivity of HSV-1 KOS, RSV strains line 19 and A2, canine coronavirus, and SARS-CoV-2 in a concentration-dependent manner (*p* < 0.05), without cytotoxic effect at all concentrations tested (CC_50_ > 200 μg/mL) (Table 1 and Appendix A). The antiviral profile of A18 was extended to inhibit the spread of HSV-1 ACV-resistant strains, B2006, and field strains, obtaining EC_50_ values similar to those observed against the KOS strain (Table 1 and Appendix A).

To further characterize the antiviral action of A18, we evaluated its effect on viral protein expression during SARS-CoV-2 infection in calu-3 cells (Figure 1A) and RSV A2 infection in A549 cells (Figure 1B) and Hep-2 cells (Appendix A), as well as on the synthesis of RNA RSV in A549 cells (Figure 1C) and Hep-2 cells (Appendix A). The number of fluorescent cells expressing RSV-F were significantly reduced in samples treated with 25 μg/mL of A18, since 93% and 96% of infected control cells expressed gF, whereas only 23% and 10% of infected and treated cells expressed gF in A549 and HEp-2 cells, respectively. Regarding SARS-CoV-2 spike protein, while 39% of infected control cells expressed Spike protein, only 7% of infected and treated cells expressed viral glycoprotein in calu-3 cells. The relative contents of viral RNA in cells infected and treated with A18 were significantly reduced in comparison to untreated infected cells at all time points p.i. (6 h, 15 h, and 24 h p.i.). Thus, these results indicated that A18 was able to inhibit the expression of some critical viral proteins and the viral RNA synthesis in the context of different virus infections.

The antiviral action of several anthocyanidins has been associated with the inhibition of the virus adsorption to the cell surface [2]. Therefore, we decided to analyze whether A18 exerts its broad-spectrum antiviral activity through this mechanism of action. As shown in Figure 2, A18 treatment during HSV-1, RSV A2, and SARS-CoV-2 binding stage resulted in a significant reduction of viral plaque formation. On the other hand, the addition of A18 during virus internalization did not reduce viral infectivity (Figure 2).

These results demonstrated that A18 had antiviral activity against HSV-1, RSV, and coronaviruses, indicating that A18 may exhibit in vitro broad-spectrum antiviral activity, probably by inhibiting viral adsorption.

### 3.2. Modulation of Cytokine Production and NF-kB Activation by A18 in RSV-Infected Epithelial and Macrophage Cell Lines

Given the significance of respiratory tract inflammation in RSV pathogenesis [20], and that it is well described the role of pro-inflammatory cytokines and NF-κB signaling pathway in their regulation [9,20], we decided to analyze the immunomodulatory properties of A18 in RSV-infected cells.

First, we measured the effect of A18 on IL-6 and IL-8 secretion in infected epithelial cells, and IL-6 and TNF-α in infected macrophages during 24 h. As a control, cells were also treated with an NF-κB inhibitor (BAY 11-7085), given that the NF-κB pathway plays a central role in mediating RSV-induced cytokine production [9,20]. As expected, IL-6, IL-8, and TNF-α production were higher in RSV A2 infected cells than in uninfected ones at 24 h p.i. [18,20,21] (Figure 3). Furthermore, A18 and BAY 11-7085 significantly reduced IL-6, IL-8, and TNF-α production in comparison to RSV-infected control cells in Hep-2, A549, and J774A.1 cells (Figure 3). Even though RSV does not multiply in J774A.1 cells in our experimental settings, it productively multiplies in A549 and Hep-2 cells. Therefore, it is possible that the decreased induction of cytokines in A549 and Hep-2 cells treated with A18 was an indirect consequence of A18-suppressed RSV replication. However, we cannot exclude the possibility that A18 was able to affect inflammatory response directly.

To unravel whether A18 could directly modulate cytokine induction in RSV-infected cells, we triggered cytokine production with non-viral stimuli, such as Toll-Like Receptors (TLR) ligands. When A549, Hep-2, and J774A.1 cells were stimulated with TLR2/6 and TLR3 ligands for 6 h, they were able to induce IL-6, IL-8, and TNF-a in these cell lines. Importantly, cytokine production was significantly inhibited by A18 and BAY 11-7085, demonstrating that A18 could directly regulate proinflammatory responses (Appendix A).

Next, we decided to investigate whether A18 could inhibit proinflammatory cytokines in infected cells by blocking the NF-κB pathway. Hence, we explored whether A18 could affect NF-κB activation by using an NF-κB-LUC reporter plasmid. We verified that RSV A2 induced NF-κB activation, and, importantly, the NF-κB signaling pathway was strongly inhibited by A18 and BAY 11-7085 in Hep-2, A549, and J774A.1 infected cells (Figure 4).

In summary, A18 proved to reduce cytokine release in RSV-infected epithelial and macrophages, probably because of the inhibition of the NF-κB signaling pathway.

### 3.3. In Vivo Evaluation of A18 Antiviral Effect

Considering that A18 exhibited broad-spectrum antiviral activities and immunomodulatory properties in vitro, we decided to investigate whether these activities were functional in vivo. For that purpose, we utilized a well-characterized model of murine pulmonary RSV infection, which is already set up in the laboratory to analyze in vivo antiviral activity of candidate drugs [20,21,22,23,24,25]. Considering that i.p. (6 mg/kg/day) and i.n. (3 mg/kg/day) administration of A18 alleviated airway hyperreactivity (AHR) in mouse models of steroid-resistant and severe asthma [14], and in view of the close association between asthma and severe RSV infection [20], we decided to conduct the experiments using these routes and doses previously reported to be active. Moreover, in uninfected mice, an i.p. dose of 6 mg/kg of A18 and an i.n. dose of 3 mg/kg of A18 had no effect on general health and behavior, as previously reported [14].

Weight loss is a quantitative measurement of the severity of RSV infection in the murine BALB/c model [2,25]. A2 and Line 19 RSV-infected mice showed an early weight reduction on day 2 p.i., (Figure 5A,B) [18,21,22,23,24]. In contrast, mice infected with both strains of RSV were treated with A18 i.p. and i.n. showed significantly less weight loss not only on day 2 p.i. but also over the entire course of the experiment when compared to untreated infected controls (Figure 5A,B). According to these results, at day 4 p.i., the time point of peak viral load in this model, RSV A2 and line 19 titers in the lung were significantly lower in A18 i.n and i.p. treated mice compared to those of untreated animals (Figure 5C).

Several cytokines have been described as important in the severity of pathophysiology and induction of AHR and mucus [18,21,25]. Interestingly, infected mice treated with A18 had significantly lower levels of IL-6, IL-4, and IL-17A than those observed in untreated infected animals (Figure 6). However, we did not observe statistical differences in IFN-g levels between animals treated or not with A18 (Figure 6).

As shown in Figure 7, RSV A2 and line 19 infections caused damage to lung tissues, as previously reported [18,21,23]. H&E staining showed alveolar walls and alveolar spaces filled with inflammatory cell infiltrates in the infected and mock-treated group. In contrast, pulmonary sections of A18-treated mice had a pattern of cell infiltration similar to those of uninfected mice (Figure 7A). When a semiquantitative histological scoring was performed, we corroborated that A18 significantly reduced RSV-induced pathological damage (Figure 7B). Thus, these results suggested that A18 reduced lung inflammation and inflammatory cell infiltration.

Taken together, these findings demonstrated that A18 restricted pulmonary RSV infection, modulated cytokine response, and attenuated lung lesions.

## 4. Discussion

Even though hospitalizations and mortality associated with COVID-19 have decreased since the rollout of the vaccines, it continues to be a serious problem among patients with SARS-CoV-2 infection at very high risk of clinical progressions, such as immunocompromised patients or patients with advanced age [26,27]. In this sense, the World Health Organization recommends the use of antivirals against SARS-CoV-2 for early treatment of COVID-19 patients at high risk of disease progression [28]. Furthermore, RSV is currently the major cause of viral bronchiolitis in young children worldwide, and it is a leading cause of morbidity and mortality in infants, the elderly, and immunocompromised individuals [29,30]. Apart from costly passive immunization, which is reserved for very high-risk infants, there is no vaccine or effective specific treatment available for RSV bronchiolitis [31]. In the case of HSV-1, it is still a leading cause of corneal disease and blindness in humans. The HSV-1-induced ocular disease occurs because of a primary infection in the corneal epithelium and the development of immunologically driven herpetic stromal keratitis (HSK). The current standard of care for HSK includes antivirals, to inactivate and prevent further viral replication, and corticosteroids to combat the immunopathological component of the disease. However, considering the rise of resistant viruses against the current antivirals and the adverse effect of corticoids, it is essential to search for new potential anti-HSV agents [32].

In addition to these current needs for effective and safe antiviral therapies, the potential for future virus outbreaks highlights the demand for broad-spectrum therapies to combat viral infections. We should initiate comprehensive drug discovery programs for viruses that have pandemic potential, focusing on prototype pathogens. These approaches should prioritize drugs that act broadly against such virus families to provide protection against novel viruses that emerge, so they are poised to be trialed in infected patients when the next pandemic emerges [1,2].

Herbal medicines have demonstrated therapeutic efficacy for symptoms of viral infection and inflammation. The bioactive components of some plant extracts endowed with antiviral activity have been identified as proanthocyanidins and anthocyanidins, among others [2]. In this sense, fruit extracts that are known to contain anthocyanidin A18 and many related compounds to A18 have antiviral activity against many viruses in vitro, such as SARS-CoV-2 [33,34,35]. Interestingly in silico finding indicates that Cyanin and Cyanidin 3-(6″-malonylglucoside) from *E. purpurea* have good inhibitory potential against SARS-CoV-2, via the inhibition of viral main protease [36]. Moreover, it has been recently reported that molecular docking and dynamics simulation of several flavonoids predict cyanidin interacts with spike protein and alters the conformation and binding-free energy suited. Thus, based on these computational predictions, the authors proposed cyanidin as an effective drug candidate against the SARS-CoV-2 spike protein [37].

In this study, we confirm the antiviral activity of A18 against SARS-CoV-2, and we report that A18 is an effective inhibitor of HSV-1, RSV, and canine coronavirus multiplication, indicating a broad-spectrum antiviral activity. Additionally, our results were consistent with earlier studies that proanthocyanidins and anthocyanidins inhibit virus attachment to the surface of target cells. In this regard, it has been proposed that the general mechanism of antiviral activity of these types of compounds could result from the natural propensity of polyphenols to bind and aggregate proteins. Thus, these protein interactions may lead to alterations of the viral capsid or envelope protein structures and functions, or masking/blocking their binding sites to cellular receptors, resulting in the inhibition of binding of the viral particles to cell receptors [2]. Therefore, this assumption is in accordance with the computational prediction that proposes that A18 may interact with the SARS-CoV-2 spike protein and alter its conformation [37].

Airway epithelial cells are the major target of RSV and SARS-CoV-2 infection in the lung, and importantly, A18 restricts line 19 and A2 RSV and SARS-CoV-2 infection in lung epithelial cells. Furthermore, A18 showed antiherpetic activity against wild-type and ACV-resistant strains in corneal epithelial cells, the target of HSV-1 multiplication in HSK. Thus, we found that A18 exhibits a broad-spectrum antiviral activity in cells implicated in viral pathogenesis.

Many viral infections, including HSV-1, RSV, and SARS-CoV-2 infection enhances the production of several cytokines, such as IL-6, IL-8, and TNF-a in epithelial cells. The expression of these cytokines promotes activation and recruitment of immune cells, such as macrophages, resulting in increased secretion of inflammatory cytokines [6,7,20]. Several reports have shown that excessive host responses induced by inflammatory mediators may be linked to hyperresponsiveness and tissue damage during virus clearance [6,20]. Thus, properly regulating viral-induced epithelial host responses can be a key step for intervention strategies. In this study, we found that A18-treated epithelial cells and macrophages produce fewer inflammatory mediators in response to RSV infection. Since the activation of the NF-κB pathway is a necessary step for the induction of many proinflammatory mediators [6,9,20], the suppressed activation of NF-kB in RSV-infected cells treated with A18 could reduce the induction of proinflammatory mediators. Importantly, we found that A18 may control cytokine and NF-κB induction independently of its antiviral activity, given that A18 affects cytokine production in Hep-2 and A549 cells stimulated with TLRs ligands. Moreover, RSV does not multiply in J774A.1 cells in our experimental settings. Consequently, the reduction of cytokine production and NF-κB activation after A18 treatment of RSV-infected J774A.1 cells, could not be due to the inhibition of viral multiplication. Thus, A18 might modulate cytokine production and NF-κB activation independently of its antiviral properties.

NF-κB pathway plays a major role in innate inflammation by mediating RSV-induced cytokine response, airway inflammation, and disease severity in vivo [9,20]. Thus, to prevent RSV-induced lung inflammation it was proposed to target the NF-κB pathway [9]. Here we found that in mice infected with RSV line 19 and A2, A18 not only significantly reduces viral titers in the lungs, but also diminishes lung injury as observed in histological studies. In addition, there was a significant decline in Th2 and Th17 cytokine gene expression in A18-treated infected mice. Considering that A18 inhibited NF-kB signaling pathway and cytokine production in infected cells in vitro, and its known immunomodulatory activity in different in vivo models [15,16], we propose that A18 may show a protective effect against pulmonary RSV infection by reducing viral infection, and by affecting the NF-κB signaling and cytokine response as well.

One issue with the current study is that A18 is an intrinsically charged molecule that could hinder it to be absorbed and distributed well. However, it has been previously reported that in several mouse models of inflammatory disease, A18 alleviates IL-17A–dependent and TH17 cell–induced inflammation [14]. Moreover, in this study, we have found that A18 exhibits antiviral activity and protection against RSV infection in mice irrespective of the doses, routes of administration, and RSV strains. Thus, we could assume that A18 may be well absorbed and transported despite being a charged molecule. Nevertheless, in the case that in vivo pharmacokinetics studies shows that A18 is not well absorbed, it could be modified or nanoencapsulated in order to improve its abortion and distribution.

## 5. Conclusions

In summary, our data provide in vitro evidence that the flavonoid cyanidin is a promising broad-spectrum antiviral drug against several viruses, including SARS-CoV-2. Our study also supports that A18 can modulate the production of cytokines and NF-κB activation in epithelial cells and macrophages regardless of its antiviral properties. Furthermore, the in vivo mouse model validates that A18 mitigates RSV infection both in terms of pulmonary pathology and lung viral load. In view of a clinical application in humans in the future, safety, tolerability, pharmacokinetics, and efficacy in other viral models in vivo, such as the SARS-CoV-2 infection model, should be assessed. Importantly, we suggest that A18 has the potential to be an advocate for the inevitable next virus outbreak and pandemic.

## Figures and Tables

**Figure 1 viruses-15-00989-f001:**
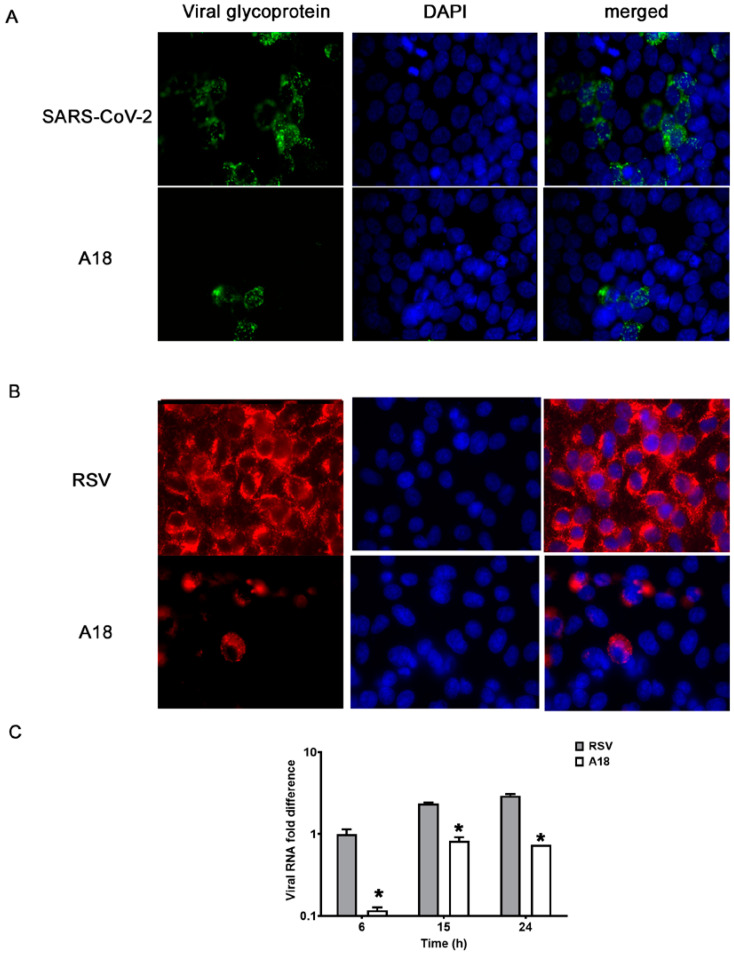
Effect of A18 on virus macromolecular synthesis. (**A**,**B**) Protein expression. IF staining was performed to detect the intracellular localization of (**A**) SARS-CoV-2 spike glycoprotein in calu-3 cells infected with SARS-CoV-2 (moi = 0.1) and (**B**) RSV gF in A549 cells infected with RSV A2 (moi = 1) and treated or not with A18 (25 mg/mL) at 24 h p.i. Magnification: 400X. (**C**) Viral RNA synthesis. Total cellular RSV RNA was extracted at the indicated times p.i. and viral RNA was quantified by qRT-PCR. The amounts of viral RNA in untreated and treated cells with A18 (25 mg/mL) in A549 cells were calculated in comparison to the content of viral RNA in untreated infected cells at 6 h p.i., defined as 1. Data represent mean ± SD for n = 3 independent experiments, performed in duplicate. * Significantly different from CV (*p*-value < 0.05).

**Figure 2 viruses-15-00989-f002:**
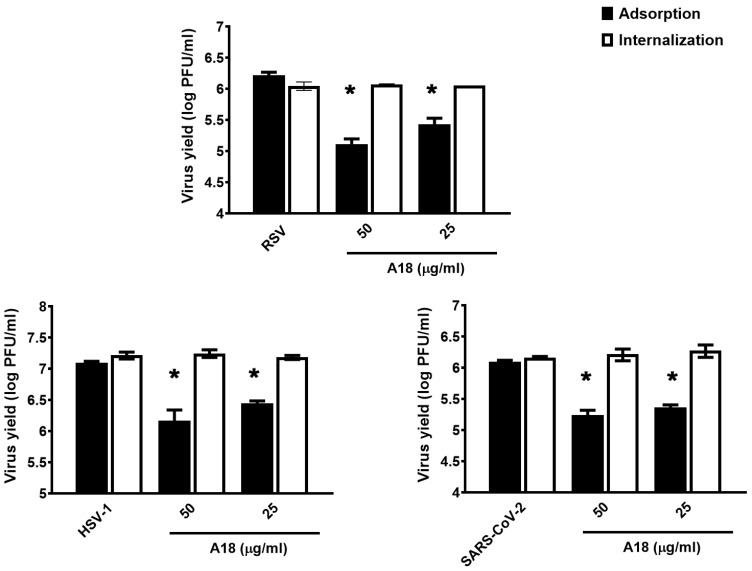
Influence of the duration of treatment with A18 on RSV infectivity. Virus adsorption: A549 cells were infected with RSV A2 (moi = 1), HCLE cells were infected with HSV-1 KOS (moi = 1), and calu-3 cells with SARS-CoV-2 (moi = 1) together with 25 μg/mL of A18 and incubated for 1 h at 4 °C. The inoculum was discarded, and cells were washed, supplemented with fresh medium, and further incubated for 24 h at 37 °C. Virus yields were collected and titrated by plaque assay. Virus internalization: A549 cells were infected with RSV A2 (moi = 1), HCLE cells were infected with HSV-1 KOS (moi = 1), and calu-3 cells with SARS-CoV-2 (moi = 1) and incubated for 1 h at 4 °C. The inoculum was discarded, and cells were washed and treated with 25 μg/mL of A18 for 2 h at 37 °C. Afterwards, cells were washed, and the non-internalized virus was inactivated with citrate buffer (pH 3) for 1 min. Cells were further incubated with fresh medium for 24 h at 37 °C. Virus yields were collected and titrated by plaque assay. Data represent mean ± SD for n = 3 independent experiments, performed in duplicate. * Significantly different from CV (*p*-value < 0.05).

**Figure 3 viruses-15-00989-f003:**
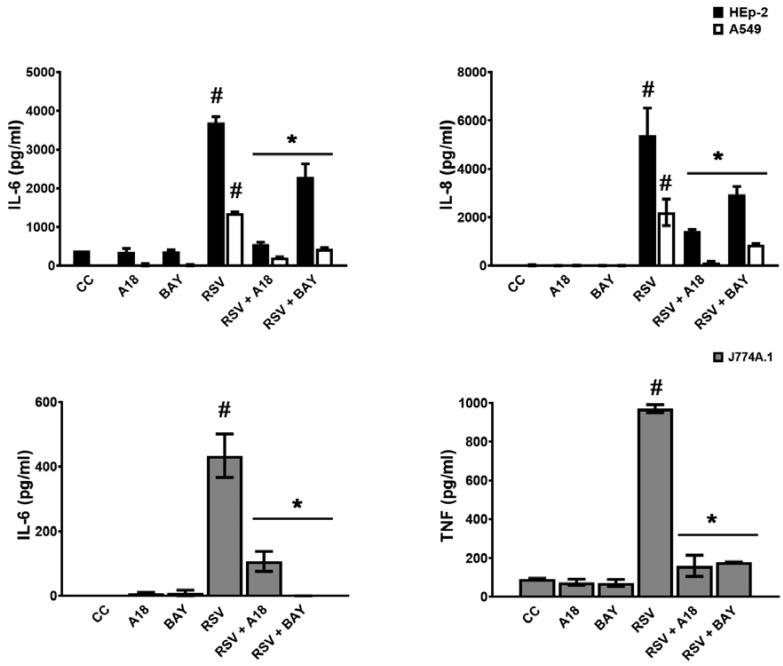
Effect of A18 on cytokine production in RSV infected cells. HEp-2, A549, and J774A.1 cells were infected with RSV A2 (moi = 1) and treated or not with A18 (25 mg/mL) and BAY 11-7085 (10 mg/mL) during 24 h. IL-6, IL-8, and TNF-α were determined by ELISA. CC: cell control (unstimulated cells). Data represent mean ± SD for n = 3 independent experiments, performed in triplicate. * Significantly different from RSV-infected cells; # Significantly different from CC (*p*-value < 0.05).

**Figure 4 viruses-15-00989-f004:**
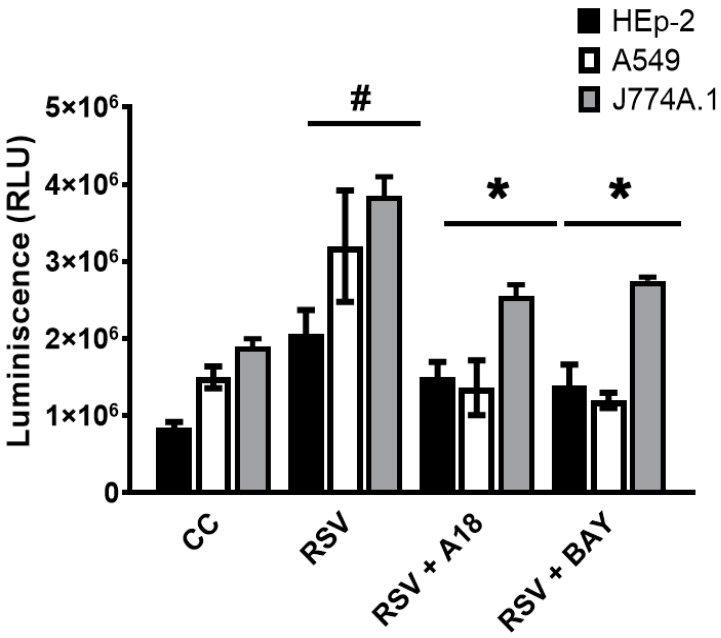
Effect of A18 on NF-κB activation in RSV infected cells. HEp-2, A549, and J774A.1 cells were transfected with 0.5 μg of NF-κB-LUC reporter vector and 0.5 μg of β-galactosidase control plasmid. After 24 h, cells were infected with RSV A2 (moi = 1) and treated or not with A18 (25 μg/mL) and BAY 11-7085 (10 mg/mL) during 6 h. Luciferase activity was measured in cell extracts, and each value was normalized to β-galactosidase activity in relative luciferase units (RLUs). CC: cell control (unstimulated cells). Data represent mean ± SD for n = 3 independent experiments, performed in duplicate. * Significantly different from RSV-infected cells; # Significantly different from CC (*p*-value < 0.05).

**Figure 5 viruses-15-00989-f005:**
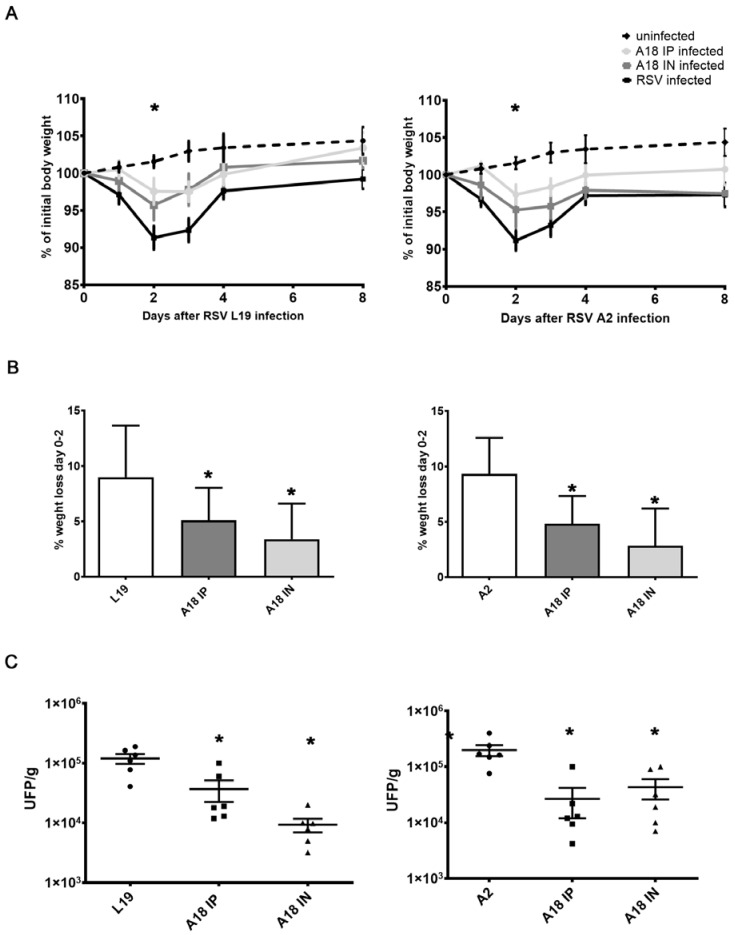
In vivo evaluation of A18 antiviral effect. Female Balb/c mice were infected with RSV line 19 and A2 (5 × 10^6^ PFU) by intranasal instillation, concomitant with 6 mg/kg of A18 i.p. or 3 mg/kg of A18 by i.n. injection on day 0. On days 1–4, all mice received further inoculations of A18 i.p. or i.n. (**A**,**B**) Weight was monitored and assessed as a percentage of starting weight. Day 0 refers to the time right before inoculation. * At day 2, values for A18-treated infected mice were significantly higher than infected mice (*p*-value < 0.05). (**C**) The animals were killed on day 4 and the lungs were used for titration of infectious virus. Data show mean ± SD from n = 6 mice/condition. * Significantly different from RSV-infected mice (*p*-value < 0.05).

**Figure 6 viruses-15-00989-f006:**
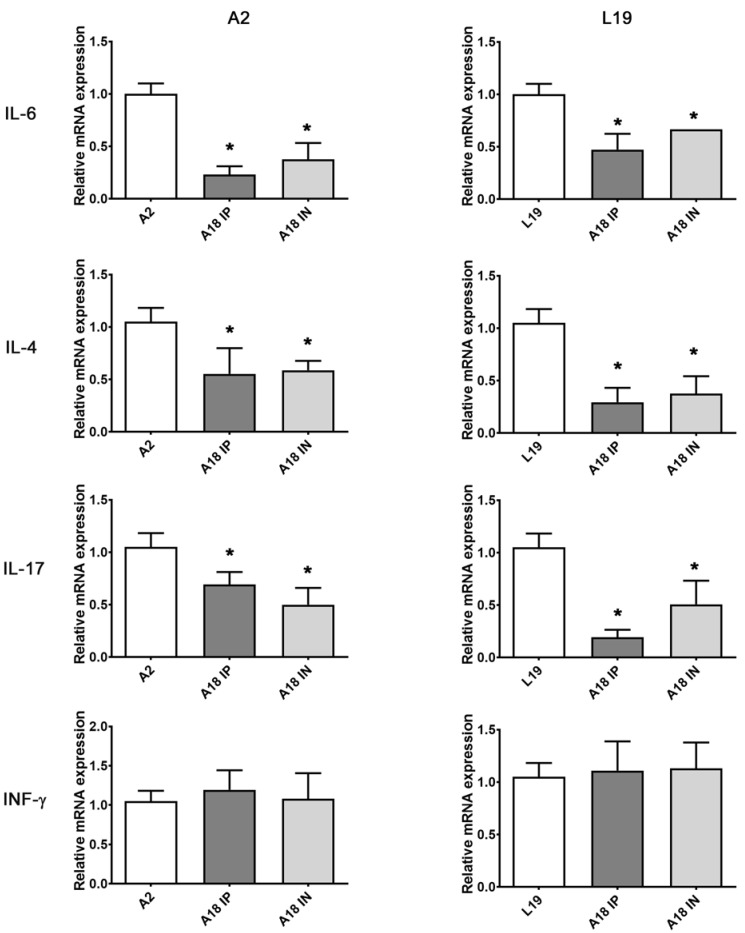
Cytokines gene expression determined by real-time PCR. Pulmonary cytokines expression was assessed on day 8 p.i. and the data were analyzed using the 2^−ΔΔCt^ formula. Actin was used as an internal for determination of gene expression. Data show mean ± SD from n = 3 mice/condition. * Significantly different from RSV-infected mice (*p*-value < 0.05).

**Figure 7 viruses-15-00989-f007:**
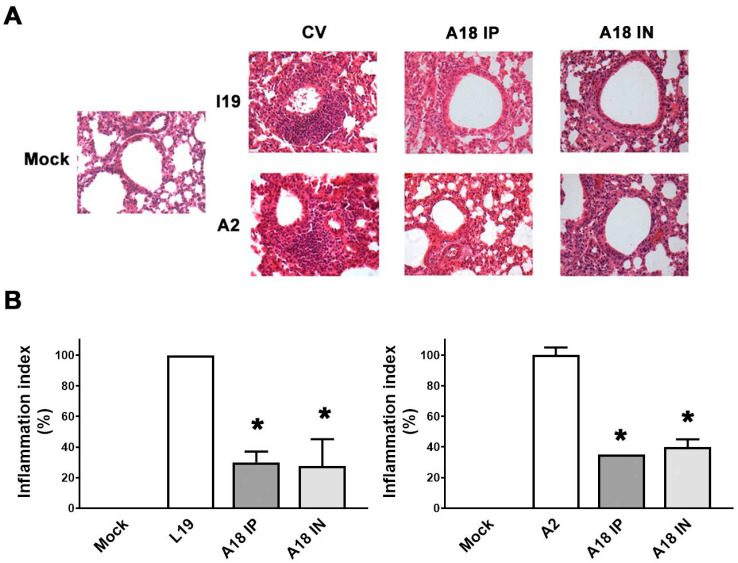
Lung histology. (**A**) Light micrographic images of pulmonary histology of H&E-stained lungs collected at day 8 p.i., shown at original magnification ×100, representative of n = 3/condition. (**B**) An index of pathologic changes in H&E slides was obtained by scoring the inflammatory infiltrate around the airways and vessels for the greatest severity. The Inflammation Index was calculated as the average of the airways’ index, and it was plotted as a percentage with respect to mock-treated infected mice, which was considered 100% of inflammation. * Significantly different from mock-treated infected mice (*p*-value < 0.05); One-way ANOVA with Tukey post-test.

**Table 1 viruses-15-00989-t001:** EC_50_ and SI of A18.

	RSV Line 19		RSV A2		HSV-1 KOS	HSV-1 Field	HSV-1 B2006	SARS-CoV-2	Canine Coronavirus
Cell Line	HEp-2	A549	HEp-2	A549	HCLE			Calu-3	CRFK
**EC_50_**	8 ± 0.1	9.8 ± 0.08	3 ± 0.06	8.5 ± 0.1	5.5± 0.1	7.6± 0.05	4.6± 0.05	8.9± 0.09	7.7± 0.1
**SI**	25	20	67	24	46	26	43	23	26

EC_50_: Effective Concentration 50. EC_50_ (μg/mL) was calculated by nonlinear regression. SI: selectivity indices (ratio CC_50_/EC_50_). SI was calculated considering as CC_50_ the maximal concentration tested (200 μg/mL). Data represent mean ± SD for n = 3 independent experiments, performed in triplicate.

## Data Availability

The data presented in this study are available on request from the corresponding author.

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
