# Peer review of "The Flavonoid Cyanidin Shows Immunomodulatory and Broad-Spectrum Antiviral Properties, Including SARS-CoV-2"

_viruses, 2023, doi:10.3390/v15040989_

Round 1

Reviewer 1 Report

The manuscript entitled “The flavonoid cyanidin shows immunomodulatory and broad spectrum antiviral properties, including SARS-CoV-2” by  Vicente et al gives an overview on the antiviral potential of cynidin on RSV, HSV-1, canine coronavirus, and SARS-CoV-2 multiplication. Additionally, authors also examined the effect of cynidin on NF-kB signaling pathway and on the production of cytokines in RSV infected epithelial and macrophages. A18 is promising antiviral agent in particular without cytotoxicity.

The manuscript is well written and will be interesting for the readers of Viruses

Figure 7 A:resolution is poor

 References need update.

Conclusion is very poor and need more elaboration with authors future perspectives.

Author Response

Reviewer 1

1) Figure 7 A:resolution is poor

Response: As requested by the reviewer, we have now provided the images with a better resolution.

2) References need update

Response: We thank the reviewer for pointing this out, and we have now updated the references.

3) Conclusion is very poor and need more elaboration with authors future perspectives

Response: We thank the reviewer for pointing this out, and we have now elaborated more deeply the future perspectives.

Reviewer 2 Report

The manuscript entitled “The flavonoid cyanidin shows immunomodulatory and broad-spectrum antiviral properties, including SARS-CoV-2” by Vicente et al reported the potential of the natural product cyanidin as a broad-spectrum antiviral agent. Despite that there are many previous reports about the antiviral potential of this natural product, the authors in the present study did a lot of in vitro and in vivo experiments to fully describe the mode of action of this molecule.

My only main concern is the poor pharmacokinetics of this compound. Being an intrinsically charged molecule hinders it to be absorbed and distributed well inside the host’s body, and several previous reports have highlighted this issue. Accordingly, how will the authors respond to this point? I think they should discuss this issue as a limitation of their study at the end of the Discussion part.

Other issues that should be addressed well by the authors:

1-The methods, particularly those describing the in vitro assays are pretty short. More details about the in vitro assays should be provided even in a supplementary file.

2- The authors pointed to a number of supplementary figures that I could not find at all. Please, revise this issue.  

3- The results section contains some information that should be described in the methodology e.g., section 3.1. Broad-spectrum antiviral activity of A18. Please, move these descriptive methods to the methodology part. 

4- “CE50 “under Table 1, I think the authors meant EC50. Please, revise. 

Author Response

Reviewer 2

1) My only main concern is the poor pharmacokinetics of this compound. Being an intrinsically charged molecule hinders it to be absorbed and distributed well inside the host’s body, and several previous reports have highlighted this issue. Accordingly, how will the authors respond to this point? I think they should discuss this issue as a limitation of their study at the end of the Discussion part.

Response: We concur with the reviewer that the mucosal absorption of many compounds is maximal at the pH at which they are mostly unionized and decreases as the degree of ionization increases. The process of drug penetration across membranes is complex, and the properties of the drug molecule that may have a bearing on membrane penetration include not only degree of ionization, but also molecular weight and size, solubility, partition coefficient, surface activity, structural isomerism, intermolecular forces, oxidation-reduction potentials, interatomic distances between functional groups, and stereochemistry. In this sense, currently there is a tremendous effort in designing effective drug delivery systems to overcome the mucus barrier and deliver therapeutics through the mucosal membrane. Promising strategies in transmucosal drug delivery rely on the development of technologies to enhance mucus permeation. One strategy is to design nanomedicines that are able to diffuse easily across mucus to reach their pharmacological targets. Nanotechnology could help overcome the imitations of conventional delivery, because they have the potential to improve the stability and solubility of encapsulated cargos, promote transport across membranes and prolong circulation times to increase safety and efficacy.  In this design process, mucus diffusion studies have an important role in the selection of the best drug candidates.

In case of A18, it has been previously reported that in several mouse models of inflammatory disease, it alleviates IL-17A–dependent and TH17 cell–induced inflammation (Liu et al, 2019). Moreover, in this study we have found that the antiviral activity and protection of A18 against RSV infection in mice is statistically significant and reproducible irrespective of the doses, routes of administration, and RSV strains. Thus, even though A18 is an intrinsically charged molecule that could hinder it to be absorbed and distributed well, we and others have found that A18 could exert anti-inflammatory and antiviral activities in mice. Nevertheless, in case in vivo pharmacokinetics studies shows that A18 is not well absorbed, it could be modified or nonencapsulated in order to improve its abortion and distribution.

Thus, we thank the reviewer for pointing this out and have now clarified this issue. We have now included the following paragraph at the end of the Discussion section of the revised manuscript:

“One issue with the current study is that A18 is an intrinsically charged molecule that could hinder it to be absorbed and distributed well. However, it has been previously reported that in several mouse models of inflammatory disease, A18 alleviates IL-17A–dependent and TH17 cell–induced inflammation. Besides, in this study, we have found that A18 exhibits antiviral activity and protection against RSV infection in mice irrespective of the doses, routes of administration, and RSV strains. Thus, we could assume that A18 may be well absorbed and transport despite is a charged molecule. Nevertheless, in case in vivo pharmacokinetics studies shows that A18 is not well absorbed, it could be modified or nonencapsulated in order to improve its abortion and distribution.”

2) The methods, particularly those describing the in vitro assays are pretty short. More details about the in vitro assays should be provided even in a supplementary file.

Response: We thank the reviewer for pointing this out, and we have now added more detail in the description of the methods. Importantly, in order to have a better description of these assays, we have also added suitable citations for each used method.

3) The authors pointed to a number of supplementary figures that I could not find at all. Please, revise this issue.  

Response:  We have previously uploaded the supplementary Figures in the same file with the Figures. In order to avoid confusions, we have now uploaded the supplementary figures as supplementary materials separately.

4) The results section contains some information that should be described in the methodology e.g., section 3.1. Broad-spectrum antiviral activity of A18. Please, move these descriptive methods to the methodology part. 

Response: As requested by the reviewer, we have now moved the descriptive methods in section 3.1 and 3.2 to the methodology part.

5) “CE50 “under Table 1, I think the authors meant EC50. Please, revise. 

Response: We thank the reviewer for pointing this out, and it has now been corrected.

Reviewer 3 Report

The manuscript entitled “The flavonoid cyanidin shows immunomodulatory and broadspectrum antiviral properties, including SARS-CoV-2” is an interesting and useful research. However, I have the following comments

1.     At the end of introduction part, write clearly the aims of your study.

2.     In the 2.1. Reagents section add the city and country for each company.

3.     Add the used positive controls in your study to validate method.

4.     Add suitable citations for each used method.

5.     In the discussion part, compare your study outcomes with previously conducted ones.

6.     The whole manuscript needs major grammar, typo and editing corrections by a native speaker.

Author Response

Reviewer 3

1) At the end of introduction part, write clearly the aims of your study.

Response: We concur with the reviewer, and we have now written more clearly the aims of the study.

2) In the 2.1. Reagents section add the city and country for each company.

Response: As requested by the reviewer, we have now added the city and country for each company.

3) Add the used positive controls in your study to validate method.

Response: Regarding the antiviral activity assay, the analysis of the percentage of inhibition was calculated with respect to the untreated virus control (positive control). Thus, in that case, a positive control was included in each antiviral assay. Nevertheless, if the reviewer refers to validating the method with a reference drug, we have now clarified in materials and methods section of the revised manuscript that we have used acyclovir against HSV-1 as a reference drug to validate the antiviral activity method. Besides, we have added the citation where we have described this method and where we have previously used acyclovir against HSV-1 as a reference drug. This method has been widely used in the laboratory to determine the antiviral activity of numerous compounds in different virus-cell systems.

In order to validate the modulation of cytokines and NF-kB activation, we have already stated that we have used BAY 11-7085 (NF-kB inhibitor) as a positive control.

4) Add suitable citations for each used method.

Response: As requested by the reviewer, we have now added suitable citations for each used method.

5) In the discussion part, compare your study outcomes with previously conducted ones.

Response: We thank with the reviewer for pointing this out, and we have now compared our study outcomes with previously conducted ones in the discussion section of the revised manuscript.

6) The whole manuscript needs major grammar, typo and editing corrections by a native speaker.

Response: As requested by the reviewer, we have now edited and corrected the manuscript by a native speaker.

Round 2

Reviewer 2 Report

The authors have addressed most of my concerns. The article is eligible to be published in its present form.

Reviewer 3 Report

The authors conducted all the required corrections and I have no more suggestions